# Rescuing SLAMF3 Expression Restores Sorafenib Response in Hepatocellular Carcinoma Cells through the Induction of Mesenchymal-to-Epithelial Transition

**DOI:** 10.3390/cancers14040910

**Published:** 2022-02-12

**Authors:** Grégory Fouquet, Constance Marié, Louison Collet, Catherine Vilpoux, Hakim Ouled-Haddou, Eric Nguyen-Khac, Jagadeesh Bayry, Mickaël Naassila, Ingrid Marcq, Hicham Bouhlal

**Affiliations:** 1Groupe de Recherche sur l’Alcool et les Pharmacodépendances INSERM UMR1247, Centre Universitaire de Recherche en Santé CURS, Université de Picardie Jules Verne, CHU Sud, 80000 Amiens, France; gregory.fouquet@u-picardie.fr (G.F.); constance.marie@u-picardie.fr (C.M.); catherine.vilpoux@u-picardie.fr (C.V.); nguyen-khac.eric@chu-amiens.fr (E.N.-K.); mickael.naassila@u-picardie.fr (M.N.); 2Laboratoire HEMATIM EA4666, Centre Universitaire de Recherche en Santé CURS, Université de Picardie Jules Verne, CHU Sud, 80000 Amiens, France; louison.collet@u-picardie.fr (L.C.); hakim.ouled-haddou@u-picardie.fr (H.O.-H.); 3Department of Hepatogastroenterology, Centre Hospitalier Universitaire Sud, 80000 Amiens, France; 4Institut National de la Santé et de la Recherche Médicale, Centre de Recherche des Cordeliers, Sorbonne Université, Université de Paris, 75006 Paris, France; jagadeesh.bayry@crc.jussieu.fr; 5Department of Biological Sciences & Engineering, Indian Institute of Technology Palakkad, Palakkad 678623, India

**Keywords:** SLAMF3, sorafenib resistance, EMT, CSCs, HCC

## Abstract

**Simple Summary:**

Sorafenib is a treatment for advanced HCC which demonstrated a poor objective response rate due to important induction of resistance. We demonstrated that induction of acquired-resistance to sorafenib in Huh-7 cell line leads to the loss of SLAMF3 expression, a tumor suppressor receptor in HCC. In these cells, the sorafenib-resistant phenotype is characterized by the increase of aggressiveness and induction of the epithelial-to-mesenchymal transition. Acquired-resistance to sorafenib induce a multipotent mesenchymal stem cells characteristic. Interestingly, SLAMF3 overexpression reversed the epithelial-to-mesenchymal transition and decreased metastatic potential in sorafenib-resistant cells through the control of ERK1/2 and mTOR signaling pathways. SLAMF3 seems to be a theranostics tools to the management of sorafenib treatment.

**Abstract:**

Background: Acquired resistance to sorafenib in hepatocellular carcinoma (HCC) patients results in poor prognosis. Epithelial-to-mesenchymal transition (EMT) is the major mechanism implicated in the resistance to sorafenib. We have reported the tumor suppressor role of SLAMF3 (signaling lymphocytic activation molecules family 3) in HCC progression and highlighted its implication in controlling the MRP-1 transporter activity. These data suggest the implication of SLAMF3 in sorafenib resistance mechanisms. Methods: We evaluated the resistance to sorafenib in Huh-7 cells treated with progressive doses (Res cells). We investigated the link between acquired resistance to sorafenib and SLAMF3 expression by flow cytometry and Western blot methods. Furthermore, we analyzed the EMT and the stem cell potential of cells resistant to sorafenib. Results: Sorafenib resistance was confirmed in Res cells by analyzing the cell viability in the presence of sorafenib. The mesenchymal transition, in Res cells, was confirmed by high migratory index and the expression of EMT antigens. Interestingly, we found that loss of SLAMF3 expression corresponded to sorafenib-resistant phenotypes. The overexpression of SLAMF3 reversed EMT, decreased metastatic potential and inhibited mTOR/ERK1/2 in Res cells. Conclusions: We propose that rescuing SLAMF3 expression in resistant cells could represent a potential therapeutic strategy to enhance sorafenib efficacy in HCC patients.

## 1. Introduction

Due to late diagnosis, only a minority of hepatocellular carcinoma (HCC) patients are eligible for curative treatment [1]. For ineligible patients, sorafenib is the main clinically approved drug for advanced HCC. Sorafenib possesses antiproliferative and antiangiogenic effects in vitro by the inhibition of Raf kinase, the vascular endothelial growth factor receptor (VEGFR) and the platelet-derived growth factor receptor (PDGFR) [2]. Clinical trials have shown that sorafenib potentially improves overall survival by 3 months in advanced HCC patients, but with an objective response rate of only 2–3%. Due to the high resistance rate, only few patients obtain a long-term benefit from sorafenib therapy [3,4].

Numerous mechanisms are proposed to be associated with the high resistance rate to sorafenib, of which epithelial-to-mesenchymal transition (EMT) has a major role [5]. Indeed, EMT characterized by the loss of epithelial cell markers and acquisition of mesenchymal cell phenotype, has been described as a physiological phenomenon in embryonic development. In addition, EMT is also associated with the poor prognosis of HCC [6]. This aggressive phenotype of HCC, acquired during the sorafenib resistance process, is associated with the loss of epithelial markers such as E-cadherin, and the simultaneous acquisition of vimentin, a major mesenchymal phenotype protein [7]. EMT is induced by the activation of several transcription factors, which act as key transcriptional repressors of epithelial-specific genes such as E-cadherin [8].

Mesenchymal stromal cells (MSCs) are undifferentiated multipotent cells, which reside in various adult human tissues and have the potential to differentiate into adipocytes, osteoblasts and other tissues of mesenchymal origin. MSCs present a spindle-like morphology and provide important migratory and invasive potential to HCC cells. MSCs are phenotypically positive for CD44, CD90, CD105, CD133 and CD73, while CD34, CD45 and HLA-DR are negative [9]. These MSC positive markers could identify cancer stem cells (CSCs). Indeed, the enrichment of CSCs, a subpopulation characterized by self-renewal and differentiation capabilities, could also contribute to chemoresistance in HCC [10]. Many studies have reported that TGF-β-induced EMT is accompanied by the acquisition of CSC properties [5,11].

CD44 has previously been described to enhance the tumorigenic capacity of HCC cells, and its expression was linked to lack of sorafenib-induced apoptosis in HCC cells [12] and a poor response to sorafenib therapy [13]. Sorafenib-resistant cells overexpress CD44 and CD133, and present high metastatic potential in vivo [14]. The expression of CD73, another CSC/MSC marker, is implicated in the sphere-forming ability of HCC cells through the PI3K-Akt pathway with increased metastatic potential [15]. CD73 is also associated with chemoresistance in many cancers, which might be explained by the induction of EMT [16].

Previously, we described the signaling lymphocytic activation molecules family 3 (SLAMF3) receptor as a tumor suppressor of HCC. SLAMF3 is highly expressed in healthy hepatocytes, but its expression is lost in HCC cells [17]. The overexpression of SLAMF3 in HCC cell line Huh-7 inhibits cell proliferation and blocks cell cycle in G2/M [17,18]. In addition, SLAMF3 inhibits ERK1/2, JNK and mTOR pathways and blocks HCC tumor progression in immunodeficient Nude mice [17]. Otherwise, SLAMF3 overexpression in HCC cells specifically inhibits the expression and activity of multidrug resistance protein-1 (MRP-1), which is overexpressed in HCC cells and involved in resistance to drugs, such as sorafenib [14,19]. These observations suggest that SLAMF3 might promote sensitivity of HCC cells to sorafenib through the MRP-1-dependent pathway.

In this sense, Dong et al. demonstrated that long-term exposure to a progressive dose of sorafenib induces novel phenotypes in HCC cells characterized by a resistance to sorafenib therapy [7]. These results and models were also confirmed by many other studies, which show the acquisition of an aggressive phenotype by HCC cells [14,20,21,22].

In this study, we explored the mechanisms by which SLAMF3 controls sorafenib resistance in HCC. We induced an in vitro resistance to sorafenib in HCC cell line Huh-7 (Res) by long-term exposure to progressive doses of sorafenib. We show that SLAMF3 expression is decreased in Res cells and corresponds with the sorafenib resistance profile. We reported that SLAMF3 overexpression rescues the sorafenib response in Res cells. In SLAMF3-overexpressing HCC cells, the induction of MET (mesenchymal epithelial transition) decreased metastatic abilities and inhibited mTOR and ERK1/2 phosphorylation. Altogether, our results suggested that SLAMF3 increases sensitivity of cancerous cells to sorafenib, and controls the metastatic mechanisms in cancerous cells by controlling the EMT.

## 2. Materials and Methods

### 2.1. Cell Lines

Human HCC-derived cell lines Huh-7 and PLC/PRF/5 (CRL-8024™) from ATCC (Molsheim, France) were maintained in Dulbecco’s Modified Eagle’s Medium (DMEM). Human HCC-derived cell lines SNU449 (CRL-2234™; ATCC) and SNU387 (CRL-2237™; ATCC) were maintained in Roswell Park Memorial Institute medium (RPMI 1640). Medium were supplemented with 10% fetal calf serum (FCS), 2 mM L-glutamine, and 100 UI/mL penicillin and 100 µg/mL streptomycin (Dutscher, Issy-les-Moulineaux, France). Cells were grown at 37 °C in a humidified incubator with 5% CO_2_.

### 2.2. Isolation and Culture of Mesenchymal Stem Cells

Umbilical cord (UC) samples from full-term normal pregnancies were obtained from the CHU Amiens Picardie maternity, after obtaining the donors agreement. Bone marrow (BM) samples were obtained from the cell therapy service of CHU Amiens Picardie. Informed consent was obtained from the donors. The study protocol was approved by the local independent ethics committee (CPP Nord Ouest II, Amiens, France; reference 22 January 2015).

Wharton jelly’s mesenchymal stem cells (WJ-MSC) and bone marrow mesenchymal stem cells (BM-MSC) were isolated following previously described protocol by Naudot et al. [23]. Medium was refreshed two times per week, and cells were harvested by trypsin once 80% confluence was reached. Cell were cultured at a density of 1 × 10^6^ cells/T175 flask at 37 °C in a humidified incubator with 5% CO_2_.

### 2.3. Establishment of Sorafenib-Resistant Cells

Sorafenib-resistant (Res) cells were established as previously described by Dong et al. [7]. Huh-7 cells were exposed to 2 µmol/L of sorafenib-tosylate (BioScience, Saint-Ouen-l’Aumône, France). This concentration was progressively increased by 0.25 µmol/L per week until maximum tolerated concentration. Res Huh-7 cells were obtained after 10 months under the optimized treatment and maintained in cultures in the presence of 4 µmol/L sorafenib. As a control, cells were exposed to equal volume of DMSO, the solvent used to dissolve sorafenib (Ctrl).

### 2.4. Cell Transfection and Plasmid Construction

Transfection of SLAMF3 was performed with pBud-SLAMF3 plasmid in Huh-7 cells, as previously described by Marcq et al. [17]. Transfection with empty pBud plasmid served as a negative control. For SLAMF3 ectopic expression, 2 × 105 of Huh-7 Ctrl or Res cells were first seeded into six-well plates for 24 h prior to transfection. Cells were transfected with 0.8 μg of plasmid DNA using the FuGENE HD Transfection Reagent Kit (Promega, Charbonnières-les-bains, France) according to the manufacturer’s instructions, and were incubated for 48 h or 72 h at 37 °C before all experiments.

### 2.5. General Experimental Design

All the following experiments are presented in Appendix A.

Sorafenib response was determined in Ctrl and Res Huh-7-derived cells in three different models. MTT assay (Sigma-Aldrich, France) was assessed in a 2D cell culture model, while the spheroid 3D model by Cell Titer-Glo 3D (Promega) was realized after three days of sorafenib exposure (0 to 50 µM). These culture models allowed us to determine IC50 to sorafenib. In the same time frame, the long-term effect of sorafenib exposure was studied by clonogenic assay. IC50 to sorafenib was also determined in PLC/PRF/5, Huh-7, SNU449 and SNU387 HCC cells.

To analyze HCC aggressiveness, we studied cell migration in a Boyden chambers assay, cell invasion in a modified-Boyden chambers, and a 3D spheroid model in geltrex extracellular matrix (Applied Biosystems).

SLAMF3 expression was quantified by Western blot and flow cytometry in Ctrl/Res cells and also in PLC/PRF/5, SNU449 and SNU387 HCC cells to determine a correlation between sorafenib response and the level of SLAMF3 expression.

Determination of mechanisms associated with the acquisition of sorafenib resistance was assessed by cell morphology and the acquisition of epithelial-to-mesenchymal transition (EMT) analysis. EMT was analyzed through the quantification of E-cadherin and vimentin by RT-qPCR, Western blot and immunofluorescence assays.

Flow cytometry was also realized to quantify surface expression of multipotent cell markers such as CD44, CD133, CD73, CD90 and CD105 (list of antibodies are presented in Appendix A). Study of cancer stemness abilities was analyzed through adipogenic differentiation. This differentiation was confirmed by Oil Red O coloration and adipogenesis-related genes (PPARγ, C/EBPα, GLUT4, Leptin and FABP4) by RT-qPCR (list of primers are presented in Appendix A).

### 2.6. Statistical Analysis

All data are expressed as mean ± SEM. The independent Student’s two-tailed t-test was used to compare Ctrl and Res cells in the different assays. Comparison between level of SLAMF3 expression and sorafenib response was performed by the Pearson correlation test. Comparison between SLAMF3 and mock conditions was realized by a two-tailed t-test. Data from IC50 and SLAMF3 expression (flow cytometry and Western blot) in epithelial-like cells and mesenchymal-like cells were assessed by two-tailed t-test. Data from sorafenib sensitivity assays and SLAMF3 overexpression assays were analyzed by an analysis of variance (ANOVA) followed by Tukey’s post-hoc tests with two factors, cell condition and sorafenib concentration or cell condition and cell transfection, respectively. For adipogenic differentiation, a one-way ANOVA test was used to compare the different conditions. Statistical analyses were performed with Sigma Plot software (version 11.0, Systat Software Inc., San Jose, CA, USA). The threshold for statistical significant was set to *p* < 0.05 for all analyses.

## 3. Results

### 3.1. Long-Term Exposure to Increased Concentrations of Sorafenib Induces Resistance and Aggressive Phenotypes in HCC Cells

We obtained sorafenib-resistant Huh-7 cells (Res) after 10 months of sorafenib exposure. Compared to Ctrl cells, Res cells exposed to sorafenib had more viability. This sorafenib resistance was confirmed by the significantly increased IC50 value in Res cells compared with the Ctrl conditions in a 2D culture model. The Res cells had fusiform morphology while cells in the Ctrl conditions were round in shape. Fifty percent of cell viability was reached in Res and Ctrl cultures in the presence of 11.40 ± 1.22 µM and 3.93 ± 0.29 µM sorafenib, respectively (*p* < 0.001; Figure 1A). The 3D culture model showed that Res cells resisted sorafenib up to 23.26 ± 0.68 μM concentration, while this was 16.27 ± 0.36 μM in Ctrl cells (*N* = 4; *p* < 0.001; Figure 1B). Furthermore, colony forming units (CFU) were drastically decreased in Ctrl cultures after 5–10 µM concentration of sorafenib. However, 15–20 µM sorafenib was necessary to limit the CFU propagation in Res cultures (*N* = 3; Figure 1C).

To confirm acquired resistance to sorafenib, we examined the apoptotic effect of sorafenib by flow cytometry-based analysis of Annexin V (AV)+/Propidium Iodide (PI)+ subpopulations. As shown in Appendix A, sorafenib exposure induced apoptosis in a dose-dependent manner in Ctrl cells. However, Res cells did not display such an effect. The percentage of AV+ or PI+ cells were similar between untreated or sorafenib-treated Res cells.

Resistance to sorafenib was accompanied by an aggressive behavior of tumor cell, as shown by significant enhancement of cell migration and cell invasion. Compared to Ctrl cells, the migratory and invasive capacities of Res cells were increased by 51.94-fold (*N* = 4; *p* < 0.001; Figure 1D) and by 105-fold (*N* = 3; *p* < 0.001; Figure 1E), respectively. The aggressiveness of HCC Res cells was also confirmed by a 3D extracellular matrix culture assay (Figure 1F). Compared with Ctrl cells, which showed a spheroidal morphology without scattering, Res cells produced many protrusions, such as filipodia, which correspond to the aggressive behavior of the Res cells. These results provide evidence that long-term exposure to sorafenib confers resistance to the drug and imparts aggressiveness to tumor cells.

### 3.2. Resistance to Sorafenib Is Associated with an Epithelial-to-Mesenchymal Transition

EMT and CSCs subpopulation have been described in many studies during the induction of sorafenib resistance [7,14]. Campbell et al. defined morphology modifications as one of the most important markers of EMT [24]. Therefore, we analyzed morphology of Res and Ctrl cells by flow cytometry and by measuring the circularity index. Analysis of size and granularity by flow cytometer indicated that Res cells were significantly smaller in size compared to Ctrl cells, and had low granularity (Appendix A). Moreover, whereas Ctrl cells presented a regular and circular morphology, Res cells had an elongated spindle-like morphology, which has a similarity to that of mesenchymal cells. The circularity index was 0.784 ± 0.008 in Ctrl cells, and was significantly decreased in Res cells (0.338 ± 0.010, *N* = 3; *n* = 138, Figure 2A). These morphological changes were proven by the loss of more than 40% of the circularity index, which strongly suggested the acquisition of a spindle-like morphology by the Res cells.

The spindle-like morphology in Res cells might be explained by the loss of adherent junctions as well as cytoskeleton modifications [25]. Then, we analyzed the expression of one of the major epithelial molecules implicated in adherent junctions, the E-cadherin, and the molecule which characterizes the cytoskeleton of mesenchymal cells, the vimentin. We determined the effect of acquired resistance to sorafenib on E-cadherin (CDH1) and vimentin (VIM) mRNA expression. CDH1 expression decreased by 166-fold in Res cells (*N* = 6; *p* = 0.002). In addition, up to 3.500-fold increased expression of VIM was observed in Res cells compared with Ctrl cells (*N* = 6; *p* = 0.001; Figure 2B).

The acquisition of EMT in Res cells was also confirmed by protein analysis. The E-cadherin protein expression was reduced by 51% in Res cells compared with Ctrl cells (*N* = 5; *p* = 0.002). The vimentin expression was significantly increased by 7-fold in Res cells (*N* = 5; *p* < 0.001; Figure 2C). We further validated EMT acquisition by immunofluorescence analysis of E-cadherin and vimentin (Figure 2D). We also performed RT-qPCR analysis of transcription factors associated with EMT. The expression of Slug (SNAI2) and Zeb2 was amplified by 179.03-fold and 13.84-fold, respectively, in Res cells (*N* = 6; *p* = 0.002, *p* = 0.008; Figure 2E). The mRNA expression of Snail (SNAI1), Twist and Zeb1 were not altered in Res cells.

### 3.3. Sorafenib Resistance Is Associated with Induction of Multipotent MSC Characteristics

The expression of some of the MSC markers are associated with multipotent and aggressive phenotypes [26]. Therefore, we checked the expression of some of the MSC markers in Ctrl and Res cells. We used human MSC derived from bone marrow (BM-MSC) and Wharton jelly-MSC (WJ-MSC) as controls (Appendix A). We showed a strongly induced expression of CD44 in Res cells. The mean fluorescence intensity (MFI) of CD44 was increased by 11-fold in Res cells. The percentage of CD44+ cells was also enhanced from 8.64 ± 2.45% in Ctrl to 87.30 ± 2.20% in Res cells (*N* = 3; *p* < 0.001; Figure 3A,B).

Furthermore, the MFI of CD73 was increased up to 9.78-fold in Res cells (*N* = 3; *p* < 0.001). Though a high percentage of Ctrl cells were positive for CD73 (81.93 ± 10.32%; *N* = 3), the expression reached nearly 100% in Res cells (99.17 ± 0.18%; *N* = 3; *p* > 0.05). No modifications in the expression of CD105, CD45, HLA-DR and CD34 were observed. When we analyzed the expression of other markers of MSC such as CD90 and CD133, we found no modifications in the MFI of CD90 while the percentage of cells positive for CD90 was significantly increased by 4.07-fold in Res cells (*N* = 3; *p* = 0.008). Of note, the percentage of CD133+ cells was decreased in Res cells compared with Ctrl cells (from 20.27 ± 3.82% in Ctrl cells to 9.17 ± 0.71% in Res cells; *N* = 3; *p* = 0.046; Figure 3B). These results suggested that sorafenib-induced resistance allows the acquisition of MSC-like multipotent phenotypes.

To confirm the multipotent potential of Res cells, we cultured cells (Ctrl vs. Res) in optimized adipogenic differentiation medium, and human BM-MSC and WJ-MSC cells were used as positive controls. Adipocyte-like cells were derived from human BM-MSC and WJ-MSC proved by Oil-Red O-positive staining (*N* = 3; Figure 3C). This differentiation was also confirmed by RT-qPCR analysis of adipogenic-related genes PPARγ, C/EBPα, GLUT4, Leptin and FABP4 (*N* = 3; Appendix A). There were no detected Oil-Red O-positive cells in Ctrl cells (*N* = 3; Figure 3C), and they were negative for the expression of adipogenic-related genes (*N* = 3; Figure 3D). Oil-Red O specifically stained adipocyte lineage in Res cells after 14 days of differentiation (*N* = 3; Figure 3C). The adipocyte lineage induction in Res cells was also confirmed by a significant induction of C/EBPα (84.70-fold compared to the baseline expression in Ctrl; *N* = 3; *p* < 0.001), GLUT4 (34.17-fold; *N* = 3; *p* < 0.001), LEP (22.59-fold; *N* = 3; *p* = 0.008), FABP4 (by 8.30-fold; *N* = 3; *p* = 0.003) and PPARγ (about 4-fold, *N* = 3; *p* < 0.001; Figure 3D).

### 3.4. SLAMF3 Expression Loss Corresponds to Sorafenib Resistance Phenotype

We analyzed the expression of SLAMF3 in the sorafenib Res Huh-7 cells. Protein analysis showed a significantly decreased expression of SLAMF3 by 37.5-fold (*N* = 4; *p* = 0.036) in Res cells (Figure 4A). Surface SLAMF3 expression was decreased by 2.77-fold in Res cells compared to Ctrl cells (*N* = 5; *p* < 0.001; Figure 4B). The MFI ratio showed that SLAMF3 expression was significantly decreased in Res cells (1.39 ± 0.08) compared to Ctrl cells (2.49 ± 0.27, *N* = 5; *p* = 0.008).

To determine a possible link between SLAMF3 expression, sorafenib resistance and tumor aggressive phenotype, we analyzed the expression of SLAMF3 in three additional HCC cell lines: PLC/PRF/5 as epithelial origin cells such as Huh-7, SNU449 and SNU387, which are mesenchymal cells.

SLAMF3 was strongly expressed in epithelial cell lines compared to its expression in mesenchymal-like cell lines. We observed two times lesser SLAMF3 expression in mesenchymal cell lines than in epithelial cell lines (*N* = 3; *p* = 0.004; Figure 4C). These results were also confirmed by the total protein analysis by Western blot (*N* = 3; *p* = 0.017; Appendix A).

Moreover, HCC epithelial cell lines showed a better response to sorafenib. The IC50 for sorafenib in PLC/PRF/5 cells was 3.19 ± 0.29 µM and was noticeably equivalent to that obtained in Huh-7 cells (3.69 ± 0.46 µM). In contrast, significantly higher IC50 was obtained in mesenchymal-like SNU449 and SNU387 cells for which the IC50 were 7.04 ± 1.08 µM and 7.58 ± 0.49 µM, respectively (*N* = 3; *p* = 0.002; Appendix A). Based on our results, we identified that a low SLAMF3 expression level indicated low susceptibility to sorafenib in HCC cell lines. This strong link suggested that high expression of SLAMF3 corresponded to the “sorafenib susceptible cells profile” and to the HCC epithelial-like cells, which are the less aggressive cells. In contrast, in HCC mesenchymal cells, low expression of SLAMF3 corresponded with the high resistance to sorafenib (R^2^ = 0.866; *p* = 0.007; Figure 4D). These data identify SLAMF3 as a receptor whose expression changes depending on the state of tumor aggressiveness, and has a capacity to strongly regulate the response to the drugs such as sorafenib.

### 3.5. SLAMF3 Expression Enhances the Sorafenib Efficacy and Inhibits Aggressiveness of HCC Res Cells

To provide evidence of the reducing effect of SLAMF3 on resistance to sorafenib, we induced the transient expression of SLAMF3 in HCC Huh-7 cells by plasmid introduction. Overexpression of SLAMF3 (Appendix A) significantly decreased HCC cell viability Appendix A/S5D). The effect of SLAMF3 overexpression on the sorafenib response was investigated by IC50 analysis (Figure 5A,B). Although SLAMF3 did not completely reverse the sorafenib efficacy in Res cells, a significant decrease in the IC50 value (more than 24%) was observed in SLAMF3-transfected cells compared with mock conditions (*N* = 4; *p* = 0.028; Figure 5C). This observation confirmed the implication of SLAMF3 in sensitizing the Res HCC cells to sorafenib.

The acquired-resistance to sorafenib was accompanied by a tumoral aggressive phenotype. Therefore, we analyzed the effect of SLAMF3 overexpression on the cell migration and invasion properties, which are the characteristics of metastatic behavior. As expected, the migratory and invasive properties were lower in Ctrl cells compared with Res cells. SLAMF3 overexpression significantly decreased the cell migration by 1.77-fold (*N* = 3; *p* < 0.001; Figure 5D) and cell invasion by 4.23-fold in Res cells (*N* = 3; *p* < 0.001; Figure 5E). These data suggest that SLAMF3 expression controls the metastatic potential in sorafenib-resistant HCC cells.

### 3.6. SLAMF3 Overexpression Reverses the EMT

In order to explain the mechanisms by which SLAMF3 rescues the sensitivity of HCC cells to sorafenib, we checked the effect of SLAMF3 overexpression on EMT. As shown in Figure 6A, the overexpression of SLAMF3 in Res cells increased the roundness of the cells. The circularity index was increased from 0.384 ± 0.014 in mock to 0.470 ± 0.014 in SLAMF3 overexpressing conditions (*N* = 3; *n* = 138; *p* < 0.001). More importantly, the effect of SLAMF3 on the circularity index was more pronounced when Res/Ctrl cells were gated for SLAMF3+ cells and analyzed by flow cytometry (from 0.338 ± 0.011 in Res cells to 0.673 ± 0.017 in SLAMF3+ Res cells; *N* = 3; *n* = 138 and 100 for Res and SLAMF3+ Res cells, respectively; *p* < 0.001; Appendix A). Moreover, SLAMF3-overexpressing cells also showed a significant increase in cell complexity by 67% and 46% in SLAMF3+ Ctrl and Res cells, respectively, compared with SLAMF3- cells (*N* = 6; *p* < 0.001 in Ctrl cells and p = 0.008 in Res cells; Appendix A).

Furthermore, overexpression of SLAMF3 significantly enhanced CDH1 expression by 3.06-fold in Ctrl cells (*N* = 6; *p* = 0.022) and by 2.88-fold in Res cells (*N* = 6; *p* = 0.032). In our experiments, SLAMF3 overexpression failed to significantly modify the expression of VIM in Ctrl and Res cells (from 1.180 ± 0.291 to 1.299 ± 0.166 in Ctrl cells and from 0.947 ± 0.066 to 0.882 ± 0.078 in Res cells; *N* = 6; *p* > 0.05; Figure 6B).

To explore the mechanisms by which SLAMF3 could reverse EMT, and as transcription factors such as ZEB2 control the expression of E-cadherin, we hypothesized that SLAMF3 could target ZEB2. In line with our hypothesis, SLAMF3 overexpression significantly decreased the ZEB2 expression by 2.30-fold in Ctrl cells (*N* = 6; *p* = 0.010) and by 2.04-fold in Res cells (*N* = 6; *p* = 0.020). Other transcription factors implicated in EMT were not significantly modified by SLAMF3 overexpression (Appendix A).

Protein analysis by Western blot demonstrated significant induction of E-cadherin in SLAMF3-transfected cells (1.35-fold and 1.73-fold induction in Ctrl and Res cells, respectively; *N* = 4; *p* = 0.026, *p* = 0.002). Moreover, we observed a significant decrease (2.66-fold) in vimentin expression only in Res cells (*N* = 4; *p* = 0.003; Figure 6C). The induction of E-cadherin in SLAMF3-transfected cells was confirmed by immunofluorescence assays (N = 3; Figure 6D) and flow cytometry analysis (*N* = 4; Figure 6E). Flow cytometry analysis confirmed the significant induction of E-Cadherin expression in Ctrl (from 5.67 ± 2.31% in mock condition to 20.53 ± 4.93% in SLAMF3 condition; *N* = 4; *p* = 0.004) and Res cells overexpressing SLAMF3 (from 2.18 ± 0.09% in mock condition to 14.25 ± 0.50% in SLAMF3 condition; *N* = 4; *p* = 0.018; Appendix A).

Of note, when we analyzed only the cells which expressed SLAMF3 (SLAMF3+ cells) (Figure 6E), we observed an important high expression of E-Cadherin with 90.56 ± 7.06% in SLAMF3+ Ctrl cells and 86.45 ± 7.70% in SLAMF3+ Res cells (*N* = 4; *p* < 0.001; Figure 6E). Despite its effect on E-cadherin and vimentin expression, SLAMF3 overexpression failed to inhibit the MSC phenotype in Res cells (Appendix A).

### 3.7. SLAMF3-Induced Cell Signaling in Sorafenib-Resistant Cells

Many cell signaling events are reported to be linked to Sorafenib resistance, such as PI3K/Akt/mTOR [27], GSK-3β/β-Catenin [28], Raf/MEK1/2/ERK1/2 [29], NF-κB [30] and LKB1/AMPK [31] pathways. First, we aimed at establishing the activation state of main signaling pathways in HCC cells following the acquisition of resistance to sorafenib. We show that acquired resistance to sorafenib activated mTOR, ERK1/2 and LKB1/AMPK pathways in Res cells. Indeed, we observed a significant increase (2.20-fold) in the phosphorylation of mTOR (Ser2448) in Res cells (*N* = 5; *p* = 0.004). In addition, phosphorylation of ERK1/2 was increased up to 15-fold in Res cells compared with Ctrl cells (*N* = 5; *p* < 0.001). For the LKB1/AMPK pathway, phosphorylation of LKB1 (Ser428) and of AMPK (Thr172) were decreased by 30 and 50%, respectively, in Res cells (*N* = 5; *p* < 0.05). Despite the fact that the activation of PI3K was majored by 1.60-fold and 2.05-fold on P85/P110 in Res cells, respectively, we did not observe any increase in Akt (Ser473) phosphorylation as previously described by Chen et al. [27]. Importantly, we observed a significant decline in the β-catenin expression in Res cells (*N* = 4; *p* = 0.043), whereas NF-κB remained unmodified in nuclear fraction (Figure 7B,C).

We then analyzed the effect of SLAMF3 overexpression on signaling pathways in sorafenib-resistant HCC cells. We observed a significant decrease in mTOR phosphorylation (2.52-fold) in SLAMF3 overexpressing Res cells compared to mock cells (*N* = 3; *p* = 0.009). Overexpression of SLAMF3 also significantly decreased ERK1/2 phosphorylation by 1.46-fold (*N* = 3; *p* = 0.010). As previously described by Marcq et al. [17], SLAMF3 expression did not modify the expression of PI3K or phosphorylated AKT (Ser473). SLAMF3 overexpression, however, did not modify LKB1/AMPK, GSK-3β/β-catenin and NF-κB pathways (*N* = 3; *p* > 0.05; Figure 7B,D).

Altogether, our results provide evidence on the implication of SLAMF3 to control main cell signaling pathways involved in tumoral neoplasic mechanisms and aggressiveness. Based on these data, we proposed a model of SLAMF3 implication in transfected Res cells (Figure 7A) and confirmed the implication of SLAMF3 in the susceptibility of HCC cells to sorafenib.

## 4. Discussion

Previously, we have proposed that SLAMF3 overexpression in cancerous cells could represent a potential therapeutic strategy to improve the sensitivity of resistant cells to the drugs of HCC patients. We reported that a high expression of SLAMF3 inhibits the efflux of molecules, which would sensitize cancerous cells to the cumulated drug in the intracellular compartment [19]. Herein, we demonstrated a decreased expression of SLAMF3 in sorafenib-resistant HCC cells compared with Ctrl cells. We provided evidence that the loss of SLAMF3 expression increases HCC aggressiveness and resistance to one of the best molecules currently used in the management of patients with HCC, sorafenib. Our results confirm the potential interest of increasing SLAMF3 expression to improve the sensitivity of resistant cells to cancer drugs. Importantly, the low expression of SLAMF3 in cancerous and drug-resistant tissue may be used as a biological marker and a theranostic tool to predict the aggressiveness behavior response to drugs. Taken together with anterior results, we propose the rescue of SLAMF3 expression as one important therapeutic strategy to control the cancerous cells proliferation, and thus the regression of tumoral foci.

We also report that a deficit in SLAMF3 expression is strongly associated with decreased sensitivity to sorafenib in different HCC cell lines. These results validate the previous data that has shown the inhibition of the activity of MRP-1, one of the drug resistance proteins, in the presence of high SLAMF3 expression [17,19].

Sorafenib is the standard first-line treatment for advanced HCC, and has significantly improved global survival of patients with advanced HCC. Unfortunately, clinical evidence confirms that many patients become resistant to sorafenib treatment. In these patients who acquired resistance to sorafenib treatment, carrying on with treatment has shown limited benefits on global survival. Many studies have described the acquisition of sorafenib resistance, in vitro, following long-term exposure to the drug [14,21,22]. Chow et al. demonstrated the enhancement of a metastatic potential of HCC cells resistant to sorafenib [14]. Herein, we mimicked resistance to sorafenib by exposing epithelial lineage cancerous Huh-7 cell line to progressive high doses of sorafenib. After 10 months of prolonged drug exposure, we obtained cells with high viability index in the presence of high doses of sorafenib.

This resistance to sorafenib was accompanied by morphology changes and the acquisition of metastatic aggressiveness behavior. Exposure to high concentrations of sorafenib rendered the cancerous cells more fusiform with a decreased expression of molecules involved in intercellular cohesion, such as E-cadherin, well known as a growth and invasion suppressor. This expression reflects an aggressive metastatic phenotype in sorafenib-resistant cells. These results are in the same direction as the acquisition of a potential to migrate, and hence indicate the higher invasive potential of Res cells than Ctrl cells [32]. Interestingly, the restoration of SLAMF3 in HCC cells reduced cell migration along with the rearrangement of cytoskeletal elements. In addition, SLAMF3 expression inhibits the vimentin expression in Res cells, which characterizes the cytoskeleton of mesenchymal cells. Taken together, SLAMF3 controls the metastatic behavior by controlling the migratory potential of cancerous cells through maintaining adherence expression molecules and guaranteeing tissue adhesion and integrity.

The loss of E-cadherin, which belongs to the large cadherin family is widely known as the hallmark of EMT. E-cadherin signaling plays an important role in HCC initiation and progression [1]. The EMT represents an aggressive HCC phenotype and is considered to be a strong signal of tumor progression and metastasis [33]. We found that the sorafenib-resistant cells are more prone for EMT than Ctrl cells. Moreover, sorafenib-resistant HCC cells presented similar phenotypes and stemness as that of mesenchymal cells obtained from human bone morrow and umbilical cord as control. We confirmed this by obtaining adipogenic-like lineage under appropriate conditions derived from HCC-sorafenib-resistant cells. The observed MSC positive markers are also known to identify CSCs. Indeed, enrichment of CSCs, a subpopulation characterized by self-renewal and differentiation capabilities, might also contribute to chemoresistance in HCC [10]. This EMT is accompanied by the acquisition of stem cell properties such as CD44 expression. The CD44 expression is associated with the loss of sorafenib-induced apoptosis in HCC [12]. Our results are corroborated by the fact that CD44 has been reported to enhance the tumorigenic capacity of HCC cells, and its expression was correlated to poor response to sorafenib in HCC patients [13]. Tumor cells undergoing EMT could migrate to distant sites and become metastatic. EMT involves signaling by several transcription factors [34]. Here, we highlighted Zeb2 as the major transcription factor implicated in the EMT through the regulation of E-cadherin expression by SLAMF3. Based on our data, we proposed a molecular model establishing a link between the acquired resistance to sorafenib, EMT and a low expression of SLAMF3, which correspond to the aggressive profile of HCC cells.

Many cell signaling events are reported to be linked to sorafenib resistance. Herein, we show that acquired resistance to sorafenib activated mTOR, ERK1/2 and LKB1/AMPK pathways in Res cells. We observed a significant decline in nuclear β-catenin expression in Res cells compared to Ctrl cells, whereas NF-κB remained unmodified. More importantly, SLAMF3 overexpression significantly decreased the mTOR and ERK1/2 phosphorylation in Res cells. As previously described by Marcq et al. [17], SLAMF3 expression did not modify the expression of PI3K or phosphorylated AKT, or the LKB1/AMPK, GSK-3β/β-catenin and NF-κB pathways in Ctrl and Res cells. Altogether, our results propose a model of SLAMF3 implication in transfected Res cells (Figure 7A) and confirm the implication of SLAMF3 in the sorafenib response of HCC cells. However, additional experiments are required to validate the role of SLAMF3 in the inhibition of HCC aggressiveness and to identify additional SLAMF3 molecular partners implicated in cancer control. Furthermore, the analysis of SLAMF3 expression during different stages of HCC might provide a pointer towards the utility of SLAMF3 as a potential marker of HCC progression. Finally, our study is very relevant and was carried out on cells of a single Huh-7 cell line. Further studies would be necessary to be able to generalize our observation to other HCC cells and cancers.

## 5. Conclusions

Our results suggest that SLAMF3 expression is a target of interest to decrease the aggressiveness of cancerous cells. The rescue of SLAMF3 expression could be one of the potent therapeutic strategies to control tumor progression. The pathophysiological mechanisms that induce the repression of SLAMF3 in HCC cells remain unknown, and additional studies are needed to identify the molecular partners of SLAMF3 to elucidate the mechanisms implicated in its tumor-suppressing functions.

## Figures and Tables

**Figure 1 cancers-14-00910-f001:**
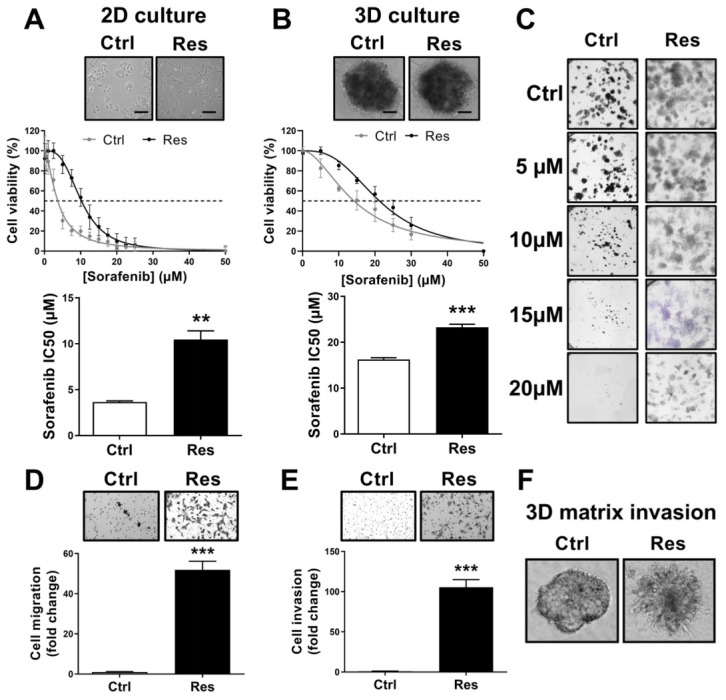
Long-term exposure to sorafenib enhances sorafenib resistance in human HCC cells and aggressive phenotypes. (**A**,**B**) Ctrl and Res cells were cultured in 2D culture conditions (**A**), or 3D culture conditions (**B**), at 0–50 µM concentrations of sorafenib. MTT (**A**) or cell-titer 3D (**B**) were performed 72 h post sorafenib treatment. Results are represented from six independent experiments and the corresponding mean sorafenib IC50 are presented (*N* = 6; ** *p* < 0.01, *** *p* < 0.001). (**C**) Colony formation of Ctrl and Res cells after two weeks of treatment with a single dose of 0–20 µM sorafenib. Clones were stained in Crystal violet and photographed (magnification ×10). Images represent one of the three independent experiments. (**D**,**E**) Migration and invasion properties of Ctrl and Res cells as analyzed by transwell migration (**D**), and invasion assays (**E**). Images of one of the representative experiments and values are presented (*N* = 4; *** *p* < 0.001). (**F**) Spheroids morphology analysis of Ctrl and Res cells in 3D extracellular matrix culture assay. Images (magnification ×100) of one of the three experiments are presented.

**Figure 2 cancers-14-00910-f002:**
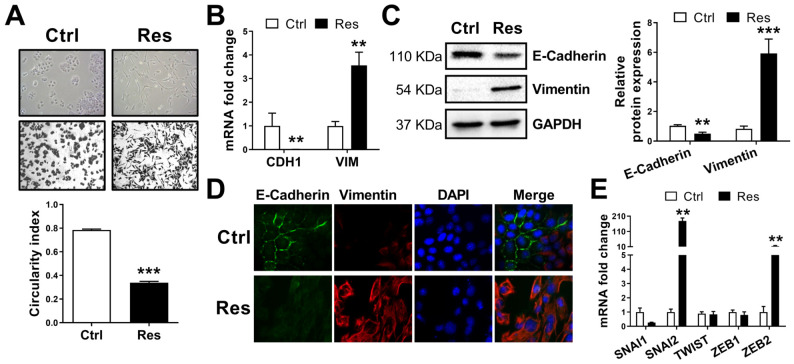
Resistance to Sorafenib in human HCC cells is associated with an epithelial to mesenchymal transition. (**A**) Photographs of Ctrl and Res cells (magnification ×). Cell morphology was analyzed by measuring the circularity index using ImageJ software (*N* = 3; *n* = 138; *** *p* < 0.001). (**B**–**D**) Effect of acquired resistance to sorafenib on the expression of EMT markers E-cadherin and vimentin as analyzed by RT-qPCR (CDH1 and VIM mRNA, *N* = 6; ** *p* < 0.01), Western blot (representative blots and densitometry analysis from 4 experiments, ** *p <* 0.01, *** *p* < 0.001 vs respective Ctrl, densitometry ratio of each band of Western blots are mentioned under bands), and immunofluorescence assays (images (magnification ×400) from the one of the three independent experiments). (**E**) Expression levels of five transcription factors associated with EMT as analyzed by RT-qPCR (*N* = 6; ** *p* < 0.01). Full pictures of the Western blots shown in Appendix A.

**Figure 3 cancers-14-00910-f003:**
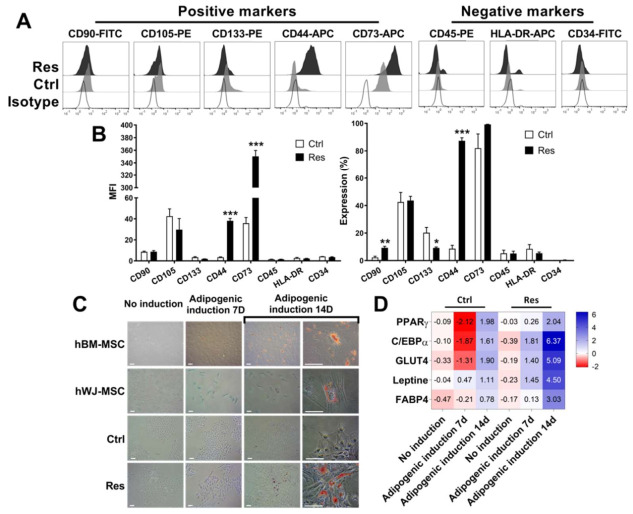
Sorafenib resistance is associated with the induction of multipotent MSC characteristics. (**A**,**B**) Flow cytometry analyses of MSC positive markers CD90, CD105, CD133, CD44 and CD73, and MSC negative markers CD45, HLA-DR and CD34 on Ctrl and Res HCC cells. Representative plots (**A**) and mean fluorescence intensity (MFI) and percentage positive cells (**B**) from three independent experiments are presented here (* *p* < 0.05, ** *p* < 0.01 and *** *p* < 0.001). (**C**,**D**) Ctrl, Res and hMSC cells were exposed to adipogenic differentiation medium for 7 or 14 days. (**C**) Oil-Red O staining of cells cultured under different conditions. Representative of one of the three independent experiments. Images are photographed at magnification ×100 and ×400 (**D**) Fold changes in the expression of adipogenic-related genes represented as a heat map.

**Figure 4 cancers-14-00910-f004:**
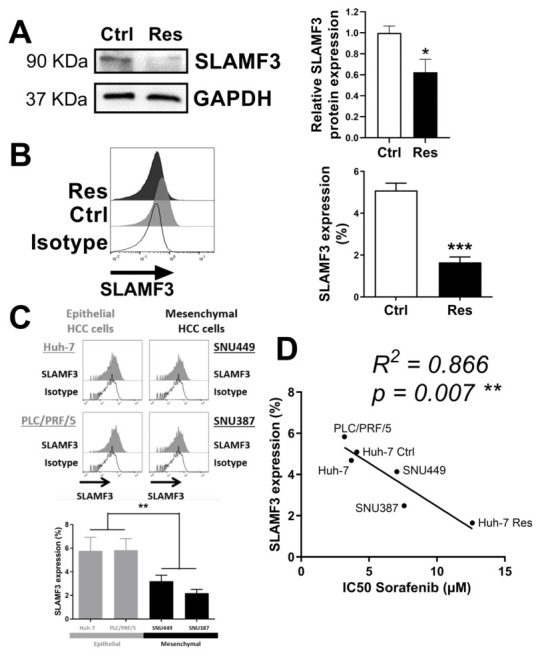
SLAMF3 expression is inversely correlated to sorafenib resistance. (**A**,**B**) The expression of SLAMF3 was analyzed by Western blot (**A**) and flow cytometry (**B**) in Ctrl and Res cells. Representative images/plots and quantification are presented (* *p* < 0.05, *** *p* < 0.001, densitometry ratio of each band of Western blots are mentioned under bands). (**C**) Flow cytometry analysis of SLAMF3 expression in four HCC cell lines. Representative plots and quantification are presented (** *p* < 0.01 epithelial cell lines vs mesenchymal cell lines). (**D**) Correlation between SLAMF3 expression and sorafenib IC50 (** *p* < 0.01; R2 = 0.866). Full pictures of the Western blots shown in Appendix A.

**Figure 5 cancers-14-00910-f005:**
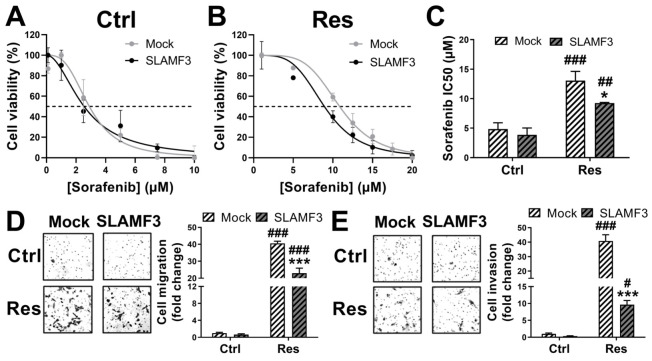
SLAMF3 expression enhances sorafenib efficacy and inhibits aggressiveness of Res cells. Ctrl and Res cells were transfected with SLAMF3 plasmid (SLAMF3) or empty plasmid (mock). The cells were treated with sorafenib by using the concentrations selected based on the results represented in Figure 1A. (**A**) Ctrl and (**B**) Res cells were cultured with 0–10 µM or 0–20 µM sorafenib, respectively, and MTT assay was performed 72 h after treatment. Results are represented as mean ± SEM of four independent experiments. (**C**) IC50 values of sorafenib in mock and SLAMF3-transfected conditions (* *p* < 0.05 vs respective mock condition; ## *p* < 0.01 and ### *p* < 0.001 vs respective Ctrl condition). (**D**,**E**) Migration and invasion properties of mock and SLAMF3-transfected HCC cells are analyzed by transwell migration (**D**) and invasion assays (**E**). Representative images and mean ± SEM are presented (*** *p* < 0.001 vs respective mock condition; # *p* < 0.05, ### *p* < 0.001 vs respective Ctrl condition).

**Figure 6 cancers-14-00910-f006:**
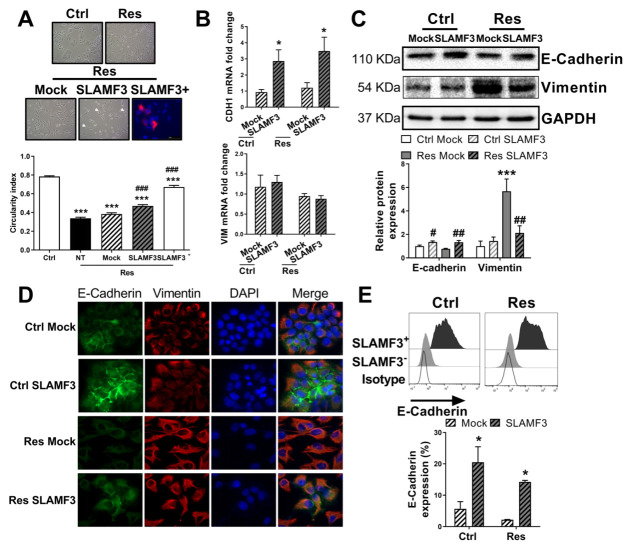
SLAMF3 overexpression promotes mesenchymal-to-epithelial transition (MET). Effects of SLAMF3 on MET were assessed 48 h post-transfection with SLAMF3 plasmid (SLAMF3) or empty plasmid (mock). (**A**) Images of cells in culture conditions or immunofluorescence staining for SLAMF3. Circularity index was analyzed by ImageJ software (*N* = 3; *n* = 138 and *n* = 100 for Res SLAMF3+ cells; *** *p* > 0.001 vs Ctrl cells; ### *p* < 0.001 vs untransfected Res cells). (**B**) mRNA expression of CDH1 and VIM quantified by RT-qPCR (*N* = 6; * *p* < 0.05 vs mock condition). (**C**,**D**) E-Cadherin and vimentin protein expression was analyzed by (**C**) Western blot (representative blots and densitometry analysis; *** *p* < 0.001 vs respective Ctrl condition; # *p* < 0.05 and ## *p* < 0.01 vs respective mock condition, densitometry ratio of each band of Western blots are mentioned under bands), and (**D**) immunofluorescence assays. Images (magnification ×400) represented one of the three independent experiments. (**E**) Flow cytometry analysis of the surface expression of E-cadherin in SLAMF3- and SLAMF3+ cells. Representative plot overlays and mean ± SEM are presented. (* *p* < 0.05 vs respective SLAMF3- cells). Full pictures of the Western blots shown in Appendix A.

**Figure 7 cancers-14-00910-f007:**
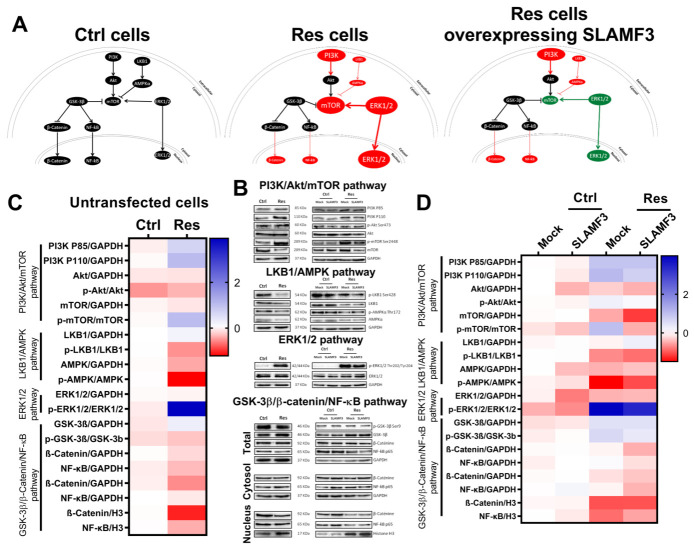
SLAMF3 reverses the cell signaling induced by long-term sorafenib exposure. (**A**) Model of cell signaling status in Ctrl, Res and SLAMF3+ Res cells. (**B**) Protein expression was analyzed by Western blot 48 h post-transfection with SLAMF3 (SLAMF3) or empty (mock) plasmid or in untransfected cells. The activation levels of PI3K (P85/P110 Thr467/Tyr199), Akt (Ser473), mTOR (Ser2448), LKB1 (Ser428), AMPK (Thr172), ERK1/2 (Thr202/Tyr204), GSK-3β (Ser9) and the expression levels of β-catenin and the p65 subunit of NF-κB are represented. GAPDH was used as a protein loading control. Densitometry ratio of each band of Western blots are mentioned under bands. (**C**) heat map representing the activation level and protein expression of various signaling molecules in Ctrl and Res cells. (**D**) Heat map representing the activation level and protein expression of various signaling molecules in Ctrl and Res cells transfected with SLAMF3 plasmid (SLAMF3) or empty plasmid (mock). Full pictures of the Western blots shown in Appendix A.

## Data Availability

The data presented in this study are available in article and Appendix A.

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
