# Peer review of "Rescuing SLAMF3 Expression Restores Sorafenib Response in Hepatocellular Carcinoma Cells through the Induction of Mesenchymal-to-Epithelial Transition"

_cancers, 2022, doi:10.3390/cancers14040910_

Round 1

Reviewer 1 Report

In the current paper entitled “Rescuing SLAMF3 expression restores sorafenib response in hepatocellular carcinoma cells through the induction of mesenchymal-to-epithelial transition”, the authors analysed the role of SLAMF3 protein in sorafenib-resistant HCC cell line.

This is an interesting piece of research, based on a relevant hypothesis, very well-written, with a well-designed experimental approach and with results well-presented and discussed. interest. The purpose of the study can pave the way to a new therapeutic strategy in the case of target therapy resistance in HCC. Only some comments are provided to improve the impact and relevance of the study.

How can the authors explain the difference in vimentin expression observed between HuH-7 cells basal condition (figure 2) and HuH-7 control used in SLAMF3 transfected cells (figure 6)?

Generally speaking, the authors observed the role of SLAMF3 in only one cell line: HuH-7. Nevertheless, the authors described all the observed effects in the Discussion section as a generalized effect in HCC without indicating that it was cell line-specific. HuH-7 cell line originates from a hepatoma tissue, which was surgically removed from a Japanese man. PLC/PRF/5 cell line is reported by the authors to have a similar epithelial isotype and to respond to sorafenib in a superimposable manner. However, the PLC/PRF/5 cell line has a different ethnicity, being the cells originated from a hepatocarcinoma tissue, which was surgically removed from a South African man. In this sense, the authors could, on one hand, analyze the molecular changes in PLC/PRF/5 and PLC/PRF/5 Res cell lines, in case to confirm the role of SLAMF3 in another cell line, and, on the other hand, tone down the discussion section highlighting the limitation of the study which reports the results obtained on a single cell model.

It would be interesting if the authors could explore the possibility that the decreased expression of the SLAMF3 receptor could have been reported in HCC cohorts, especially in those more aggressive and/or resistant to sorafenib treatment. In silico approach in available datasets could be an option.

Author Response

Response to Reviewer 1 Comments

This is a sound paper showing that SLAM3 reverses sorafenib resistance in HCC. The science is sound but several minor questions remain. Why did SLAM3 fail to reverse the mesenchymal phenotype? The reversal of sorafenib sensitivity by SLAM3 is modest (Fig 5C). If SLAM3 is the driver for EMT won't we expect more dramatic reversal? How would the authors explain the decline in CD133 while other Mesenchymal markers are going up? As well as is there any potential explanation why different markers are behaving differently? MET – define line 97 Figure 4B bar graph – SLAM3 expression (%) - % of what?

Point 1: Why did SLAM3 fail to reverse the mesenchymal phenotype?

Response 1: We agree with the reviewer. The expression of SLAMF3 in Res cells as evidenced by flow cytometry analysis, induced by ectopic transfection, reaches 30-40% (fig S5B). In this sense, if we take into consideration that only 30-40 % of cells are positive for SLAMF3, could explain partial effect on the mesenchymal phenotype. In addition, the SLAMF3 ectopic expression was transient that may explain, in part, the partial effect on mesenchymal phenotype.

Point 2: The reversal of sorafenib sensitivity by SLAM3 is modest (Fig 5C). If SLAM3 is the driver for EMT won't we expect more dramatic reversal?

Response 2: We agree with this point. Indeed, we show that the ectopic introduction of SLAMF3 induces EMT. Indeed, the effect of sorafenib is much more pronounced on Res cells than on ctrl cultures. This effect, on the ctlr cells, could be explained by the level of expression of SLAMF3 following ectopic introduction, and which was in the range of 30-40% of the cells. In addition, we have previously demonstrated that this expression of SLAMF3 itself induces mortality (Marcq I, Plos one 2013 doi:10.1371/journal.pone.0082918). Therefore, our observation suggest that there is only a minor additive or synergistic effect between the sorafenib and the SLAMF3 effect. We could also suggest that these two effects have independent mechanisms and different signaling pathways that need to be elucidated.

The most important point is that when SLAMF3 is reintroduced into the Huh-7 cancer cell, it becomes more sensitive to sorafenib with a more pronounced effect compared to the same cancer cells that were made resistant to sorafenib.

Point 3: How would the authors explain the decline in CD133 while other Mesenchymal markers are going up?

Response 3:

Our expression of SLAMF3 is transient on Res or Ctrl cells. This partial expression induces several effects such as apoptosis, inhibition of proliferation and even partial MET transition. We further showed that SLAMF3 induces pro-apoptotic signaling pathways and downregulates constitutively-induced activation pathways such as MAPK in cancer cells (Marcq I, Plos one 2013 doi:10.1371/journal.pone.0082918). On the other hand, we have shown that SLAMF3 decreases the expression of certain proteins involved in drug resistance including the memebrs of the MRP family. Indeed, this effect was only specific on the MRP 1 protein and not on the MRP2 and 3 (Fouquet G, OT, 2016, DOI: 10.18632/oncotarget.8679). In the same line, the effect on signaling was more specific on the phosphorylation of ERK and mTOR but not on the PI3K pathway (Bouhlal H, OT, 2015 doi: 10.18632/oncotarget.6954). All these data suggest that the effect of SLAMF3 expression would be specific to certain pathways. Our present study shows an effect of SLAMF3 expression on EMT induction but induces only certain markers, in particular, the most studied CD133 as the most CSC “cancer stem cells” marker, and not others. Many studies have shown that CD133+ cells have stemness properties such as self-renewal, differentiation ability, high proliferation and also an ability to form tumors in xenografts. In hepatocellular carcinoma cells, it has been reported that CD133 positive cells possess high capacity for tumorigenicity (YinS,. Int J Cancer 2007; 120: 1444-1450).

Considerable evidence correlate the acquisition of CSC traits with the EMT. As described, it is the process by which transformed epithelial cells can acquire the mesenchymal phenotype as well as the ability to migrate, invade, resist apoptosis, and disseminate (Thiery JP, Epithelial–mesenchymal transitions in development and disease. Cell 2009; 139: 871-890).

EMT is characterized by the altered expression of cell surface markes, increased tumor formation and is thought to endow cancer cells with migratory and invasive properties. A wide variety of solid tumours that express stem/progenitor cell marker such as CD133 have been reported to have more aggressive biological behaviour, poor prognosis and high recurrence [Elena Irollo et al. CD133: to be or not to be, is this the real question?Am J Transl Res 2013;5(6):563-581). At the same time, EMT has been suggested to generate CSCs. In this scenario, it is possible to consider the expression of CD133 as a prognostic marker for high grade tumors (Pirozzi G, et al. Epithelial to mesenchymal transitino by TGF-β1 induction increases stemness characteristics in primary non small cell lung can- cer cell line. PLoS One 2011).

Experimental evidence supports the growing importance of CD133 as a principal marker of CSCs and could be a therapeutic target. Currently CD133 is used for detection of CSCs in several malignant tumors. Taken together, the expression of CD133 must be considered as the most important to characterise the SLAMF3 effect on EMT and acquisition of cancer aggressiveness behaviour. The induction of other markers was heterogeneous and do not represent the specific effect of SLAMF3.

Point 4: As well as is there any potential explanation why different markers are behaving differently?

Response 4: Please refer to response 3

Point 5: MET – define line 97 Figure 4B bar graph – SLAM3 expression (%) - % of what?

Response 5: modifications were introduced in our corrected version.

Reviewer 2 Report

This is a sound paper showing that SLAM3 reverses sorafenib resistance in HCC. The science is sound but several minor questions remain. Why did SLAM3 fail to reverse the mesenchymal phenotype? The reversal of sorafenib sensitivity by SLAM3 is modest (Fig 5C). If SLAM3 is the driver for EMT won't we expect more dramatic reversal? How would the authors explain the decline in CD133 while other Mesenchymal markers are going up? As well as is there any potential explanation why different markers are behaving differently? MET – define line 97 Figure 4B bar graph – SLAM3 expression (%) - % of what?

Author Response

Response to Reviewer 2 Comments

In the current paper entitled “Rescuing SLAMF3 expression restores sorafenib response in hepatocellular carcinoma cells through the induction of mesenchymal-to-epithelial transition”, the authors analysed the role of SLAMF3 protein in sorafenib-resistant HCC cell line.

This is an interesting piece of research, based on a relevant hypothesis, very well-written, with a well-designed experimental approach and with results well-presented and discussed. interest. The purpose of the study can pave the way to a new therapeutic strategy in the case of target therapy resistance in HCC. Only some comments are provided to improve the impact and relevance of the study.

Point 1: How can the authors explain the difference in vimentin expression observed between HuH-7 cells basal condition (figure 2) and HuH-7 control used in SLAMF3 transfected cells (figure 6)?

Response 1: We agree with the reviewer on this point that just transfection would induce the expression of vimentin. Indeed, vimentin is induced in mock cells as shown by microscopy and Western blot. This expression is more strongly induced by the SLAMF3 and is significantly inhibited in resistant cells. We could partly explain this non-specific induction of vimentin in mock cells by the state of stress of the cells by the simple transfection. Indeed, it has been shown that a state of cellular stress or inflammatory conditions could induce this expression during the EMT process (Renaud-Picard B. et al. Transplant Immunology, EMT and membrane microparticles: potential implication for bronchiolitis obliterans syndrome after lung transplantation, 2020).

Point 2: Generally speaking, the authors observed the role of SLAMF3 in only one cell line: HuH-7. Nevertheless, the authors described all the observed effects in the Discussion section as a generalized effect in HCC without indicating that it was cell line-specific. HuH-7 cell line originates from a hepatoma tissue, which was surgically removed from a Japanese man. PLC/PRF/5 cell line is reported by the authors to have a similar epithelial isotype and to respond to sorafenib in a superimposable manner. However, the PLC/PRF/5 cell line has a different ethnicity, being the cells originated from a hepatocarcinoma tissue, which was surgically removed from a South African man. In this sense, the authors could, on one hand, analyze the molecular changes in PLC/PRF/5 and PLC/PRF/5 Res cell lines, in case to confirm the role of SLAMF3 in another cell line, and, on the other hand, tone down the discussion section highlighting the limitation of the study which reports the results obtained on a single cell model.

Response 2: Indeed, our study was carried out on a single cellular model, based on the observation that it is a reliable, representative and easy-to-use model. We have also compared this line to other HCC lines on their resistance to sorafenib treatment. Indeed, the PLC/PRF-5 cells exhibit the highest level of expression, and was inversely proportional to their very high sensitivity to sorafenib. We could not obtain PLC/PRF-5 resistant cells nor could analyze the different modifications in these cells induced by SLAMF3. This is partly due to the basal fragility of the cells of this line, given the high basal level of SLAMF3.

We have modified the discussion, as suggested by the reviewer concerning the impossibility of extrapolating our results obtained on Huh-7 on all HCC cells.

Point 3: It would be interesting if the authors could explore the possibility that the decreased expression of the SLAMF3 receptor could have been reported in HCC cohorts, especially in those more aggressive and/or resistant to sorafenib treatment. In silico approach in available datasets could be an option.

Response 3: This point is crucial and we thank the reviewer for raising it. Indeed, for the ongoing other project, we constituted a cohort (n=18, 10 non-responders and 8 responders to sorafenib) of patients and analyzed their hepatic cells for the expression of SLAMF3 mRNA by RT-PCR. The definition of responders or not to sorafenib is based on the criteria defined by the clinicians' recommendations: responders = stable non-evolving tumors, non-responders = escape and progression of the tumor (follow up period of 3 months). The indisputable results show that the SLAMF3 mRNA level is very low in non-responders (n=10), compared to its, relatively, high expression in patients clinically non-progressive or responders (n= 8, Figure only for the reviewers and Editor). Our pilot study is very encouraging and prompted us to analyze the impact of SLAML3 expression in a larger cohort. Taken together, our results strongly suggest that expression of SLAMF3 could be used as a theranostic tool to define the susceptibility of cancer cells to the drugs.

Figure to reviewer (attached document)
